# When revealed after the fact, selfish intentions undermine prosocial actions in 5-year-olds

Kayley Dotson[1]*, Michael Tomasello[2,3]

1 Department of Psychology, University of Michigan, Ann Arbor, Michigan, United States of America,
2 Max Planck Institute of Evolutionary Anthropology, Leipzig, Germany, 3 Department of Psychology and Neuroscience, Duke University, Durham, North Carolina, United States of America

* kayleyd@umich.edu

## Abstract

Early in ontogeny, children show a preference for prosocial others and for those with helpful intentions. Here, we investigate how children reason about prosocial actions when a selfish intention is revealed only after a prosocial behavior has benefitted them. 5-year-old children (N = 48) played a game with a puppet who acted prosocially by letting the child take a turn in a game. After the game, the puppet revealed either a selfish intention of getting a cookie for letting the child play (Undermining condition) or a prosocial intention of wanting the child to have fun (Prosocial condition). When given a chance to reciprocate a prosocial act, children shared less when the puppet had an Undermining versus a Prosocial intention. Children did not show any condition differences in their explicit social evaluations of the puppet, however. Our results indicate that an agent's initially hidden selfish intention, revealed only after a prosocial action, negatively impacts a child's willingness to later reciprocate with that agent.

## Introduction

From early in life, infants prefer those who act prosocially [1–5]. They anticipate and prefer agents who engage in helpful versus harmful actions [6], distribute resources fairly [7–9] and comfort others in distress [10–12]. Though this preference may be weaker or develop later than initially thought [13–14], by the time children reach toddlerhood, they reflect a strong prosocial preference through their own behaviors. Across cultural contexts, infants and toddlers share information through gestures like pointing which become more socially motivated from year one to year two [15–17]. They also infer others' mental states and behave prosocially by comforting others [18–19] and providing instrumental help to achieve goals [20–23]. These prosocial actions become more intentional with age, as children help more autonomously and with less scaffolding over the second year of life [24]. Young children not only prefer prosocial others but actively respond when others fail to act prosocially. As young as 3-years-old, children report and protest others' antisocial actions [25–27]. They also punish antisocial agents, intervening just as much when violators act against them

**Data availability statement:** All relevant data are within the paper and its Supporting information files.

**Funding:** The author(s) received no specific funding for this work.

**Competing interests:** The authors have declared that no competing interests exist.

or others [28]. Thus, the preference for prosocial behaviors and against antisocial behaviors is well-documented in young children.

This early prosocial preference shapes children's emerging sense of reciprocity. Infants are not only sensitive to prosocial behaviors, but they expect prosociality to be reciprocated. As early as 15 months, infants are surprised when an agent fails to help a prior helper [29]. As children develop, this expectation translates to their own behaviors. Three-year-olds, but not two-year-olds, share more with those who have already shared with them [30], demonstrating the emergence of direct reciprocity. Children also engage in indirect reciprocity. After receiving help, 4-year-old, but not 3-year-old, children are more likely to "pay it forward" both positively and negatively to a novel agent [31–32]. Furthermore, 3.5-year-olds prefer giving to individuals who previously shared with them and to those who shared with others [33]. Therefore, by preschool age, reciprocity is guided by information about prosociality.

Although young children reciprocate based on others' prosocial behavior, these decisions may not be based purely on prosocial motivations. Indeed, reciprocity becomes increasingly strategic over development. By age five, children selectively reciprocate with partners who can reciprocate themselves [34–35]. However, this strategic reciprocity may not develop uniformly. Four- to 8-year-old children engage in negative direct reciprocity but do not engage in positive direct reciprocity until age 7 [36], suggesting that children more readily punish those who act selfishly versus reward those who act prosocially. Children also begin to use prosociality instrumentally around this time. Five-year-olds and 7-year-olds (but not 3-year-olds) are able to act generously in a selfish manner to later be selected as a play partner [37]. In some situations, the desire to reciprocate strategically may even override other social norms. Five- to 8-year-old children prioritize reciprocity over honesty and engage in cheating behavior to benefit an agent who had previously shared with them [38]. Over the first few years of life, reciprocity becomes more strategic and guided by more than others' explicit behaviors alone.

Beyond a sensitivity to others' actions, one critical factor in children's emerging reciprocity decisions may be their developing sensitivity to others' intentions. The ability to understand intentions may be rooted in evolution, with chimpanzees showing the ability to distinguish those who are unwilling versus unable to complete a task [39], though humans appear to uniquely and intrinsically prefer prosocial intentions [40]. By 10 months, infants can recognize whether an action is intentional versus accidental and whether a goal is successful versus unsuccessful [41–42]. Children also use intention information to guide their prosocial behaviors by two years of age [43]. By preschool age, children selectively avoid those with harmful or hindering intentions and reciprocate more generously to someone who intentionally benefits them [44–46]. Even when outcomes do not align with intentions, children often still prioritize prosocial intentions by age two [47] though they evaluate both intentions and outcomes in their moral evaluations [48–51]. As theory of mind develops, children's use of intention information grows more sophisticated: 5-year-olds who pass a theory of mind task attribute less blame to an accidental transgressor than 3-year-olds, who prioritize outcome [52–53]. However, in prior studies, children knew the

agent's intentions as the action occurred or beforehand, inherently tying the action and intention together. To our knowledge, no prior work explores how children respond when intention information is revealed only after they have already benefitted from the prosocial action. This raises an important question about how children reason about prosocial actions and their underlying intentions in situations of potential reciprocity.

Reciprocity decisions and social evaluations require children to balance sources of information, including benefits they may receive from others' actions and the intentions that underlie these actions. This creates a possible tension between evaluating what an agent did versus why they did so. In adults, similar tensions have been proposed in the theoretical evolutionary model the homo moralis, which demonstrates that adults reason with a balance between self-interest and a preference for moral behavior [54–55]. Exploring whether similar reasoning exists in early childhood may provide insight into how young children consider and navigate both actions and intentions in their social decision-making.

While children prefer prosocial individuals with prosocial intentions, prior work largely overlooks how they respond when a prosocial behavior occurs and an underlying intention is revealed only afterward. The present study addresses this gap by isolating intention information to after the prosocial action, allowing us to investigate how children weigh the prosocial action itself versus the agent's revealed motivation in their reasoning. We therefore ask how children reason about an agent when a self-interested motivation for a seemingly prosocial action is revealed only after the fact. Since the prosocial action has already taken place and children will have already received its benefit, children must decide how to balance the prosocial action itself and the intention underlying the action in their reciprocity decisions. Specifically, do 5-year-olds reciprocate less towards a partner who reveals a selfish intention versus one who reveals a prosocial intention? If children prioritize the prosocial intention, we predict that children would be less likely to reciprocate towards the agent with a selfish intention. However, if children prioritize the prosocial action, reciprocity decisions should be similar across conditions.

To answer this question, we conducted an online experiment with 5-year-olds. Five-year-olds are at a transitional point, wherein they have a theory of mind and can strategically engage in reciprocity but are not yet necessarily fully guided by moral or cultural norms, making them the ideal age to explore how children balance prosocial actions and their benefits versus selfish intentions [56–62]. Children were taught a novel game with a potential play partner (E2). When given the choice to play themselves or let the child play, E2 acted prosocially and let the child play the game. After playing, E2 either revealed a selfish intention (to receive a cookie much better than the game) or a prosocial intention (to want the child to have fun). Children then answered questions about reciprocity and social evaluation.

## Methods

### Participants and design

The final sample consisted of 48 5-year-old children ($M_{age}$ = 5.49 years, 23 male). A sample size of 48 was selected as it is conventional in the field and, in this case, slightly larger given that the study focused on a single age group [31,63]. Participants came from mostly middle-class and upper-middle-class families in the southeastern United States. Participants were 83% White, 10% Multiracial, 4% Latinx, 2% Asian, and 2% Black. Participants were recruited from an online university database of local families. The experiment was conducted between subjects with two conditions: (1) Undermining and (2) Prosocial. The only difference between the two conditions were the intentions behind E2's prosocial action, which were explained at the end of the experimental manipulation. In addition to the 48 children in the final sample, nine participants were excluded due to distraction during the manipulation (4), strong preference for one puppet's name (2), experimenter error (2), or failure to complete the task (1).

The study was approved by the Duke University Institutional Review Board (protocol 2023−0009). All participants completed the study over Zoom. Participants were recruited from April 2023 through February 2024. Written informed consent was obtained from participants' parents or legal guardians prior to the session, and children assented before beginning the session This study was not pre-registered.

## Procedure

Participants were randomly assigned to a condition prior to beginning the study. E1 was not blinded to condition, as she needed to know the condition in order to run the proper manipulation. E1 first greeted parents and children when they joined Zoom. Parents were then asked to reaffirm their consent and were asked to hide their child's self-view panel (hiding their video) to reduce distractions. E1 then spoke with the child and attained assent before beginning the study. E1 was the same experimenter for all sessions.

E1 then began the warmup phase. During the warmup phase, E1 acted slightly surprised and explained that there was a puppet who wanted to introduce himself. E1 asked the child if she should let in the puppet, and the puppet (E3) entered. E3 was always an elephant named Alex. E3 spoke directly with the child and asked the child their name and what they liked to do in a monotone, but pleasant voice. After the child responded, E3 said that they liked to sit. E3 then said they had to leave and said goodbye. E3 was introduced at the beginning to serve as a neutral, somewhat positive agent. E3 would later be included as a potential play partner in our dependent measures. This concluded the warmup phase.

Then, the test phase began. E1 then told the child that she had a game she wanted to teach to the child called Daxing. She explained that her things for Daxing were in the other room and that she would turn off her camera and microphone but would return in just a moment with the game.

E1 then played a pre-recorded video of the introduction to E2, the introduction to the game, the game, and the experimental manipulation. These videos were pre-recorded to reduce any individual discrepancies in the experimental manipulation or game. E1 was the same across all participants, and E1 always wore the same sweatshirt to maintain consistency with the video.

When the video began, E1 "heard someone" and then another puppet (E2) entered and asked if they could play the game as well. E2 was always a giraffe named Sam. E2 then introduced themselves to the child. A few seconds then passed with no other movement in case the child introduced themselves as well or wanted to say hi. E1 then explained the game to both the child and E2. The game consisted of a ball and two buckets that made fun noises when the ball was dropped in them. E1 then explained that only one of either the child or E2 could play. E1 then let E2 decide who would get to play.

E2 then explained how much they wanted to play but that the child was their new friend. After thinking for a few seconds, E2 decided to let the child play. E2 always chose to act prosocially and let the child play. The child then played the game with E1. Children were given the opportunity to play by saying "now," at which point E1 dropped a ball in a bucket that made a fun noise. The child did this four times. Videos were timed to give children enough time to respond. Children who did not participate by saying "now" were excluded.

After four rounds of the game, E1 sat back and E2 returned excitedly. E2 asked if the game was over, which E1 confirmed. E2 then said they hope the child had fun playing the game. In the Prosocial condition, E2 said, "I only let you play because I wanted you to have fun! You had fun! I'm so glad I let you play daxing. Yay!" However, in the Undermining condition, E2 says, "I only let you play because I wanted to get a cookie instead! I get a cookie! This is so much better than daxing. Yay!" Immediately after this, E2 said they had to go, said bye, and left. E1 then said that they should go back to the other room now. This concluded the pre-recorded video and the test phase. Children then answered questions about reciprocity and social evaluation. See Fig 1 for a graphical representation of the two conditions. The full procedure script can be found in S2 File.

## Measures

Our main dependent variable was (1) reciprocity. We also measured (2) social evaluation through three measures: (a) liking, (b) partner choice, and (c) social bonding. Children always completed the measures in the same order: liking, partner choice, reciprocity, and social bonding. This order was set due to the structure of the tasks and the materials. First, children viewed a shared screen which displayed the liking and partner questions. Next, the experimenter ended

**Fig 1. Graphic depicting the Undermining condition versus the Prosocial condition manipulation.**

the shared screen display and retrieved the bouncy ball used for the reciprocity measure. Finally, children completed the social bonding measure. Because this measure required children to wait, it was administered last so that children could immediately transition to a fun, unrelated game before ending the study, minimizing any frustration or disengagement from the participant.

**Reciprocity.** Reciprocity was measured through a prize allocation task, where children decided if they wanted to give E2 a prize that E1 found. E1 "found" a bouncy ball near her computer and excitedly showed the child. E1 then asked the child if they thought E1 should keep the ball or give it to E2. Participants chose whether E1 kept a toy (colorful bouncy ball) or gave it to E2. Reciprocity was a forced choice of telling E1 to either (1) keep the toy (no reciprocity) or give the toy to E2 (reciprocity).

**Social evaluation.** Social evaluation was measured through a liking score, partner choice, and social bonding. For liking, participants answered how much they liked E2 on a 7-point smiley face Likert scale [1 = *does not like at all,* 7 = *likes a lot*]. For forced partner choice, participants chose who they would invite to their own playdate between E2 and E3 (forced choice). Finally, social bonding was measured through a task assessing children's willingness to stay and wait for subsequent interaction with E2. Participants were told that E2 may be coming back soon and that there was a game to play. They were told that we could go ahead and play the game without E2, or that they could wait for E2 so that they could all play the game. If the participant initially answered that they wanted to wait, they were then asked at 10-second intervals if they would like to continue to wait. Participants could wait for a maximum of seven prompts (or 60 total seconds). If a participant reached five prompts, the experimenter expressed uncertainty that E2 may not be returning, and the participant was asked if they want to keep waiting or play the game. If a participant reached six prompts, the experimenter suggested that E2 may not be returning, and the participant was asked if they want to keep waiting or play the game. If a participant reached seven prompts, the experimenter said that E2 must not be returning and that they should proceed to the game. This measure was recorded as 1–7, with 1 being did not wait at all and 7 being waited 60 seconds (all of the prompts). After waiting, children immediately played the game. This game consisted of E1 saying a pre-set list of objects and the child saying whether that object was big or little as fast as they could (e.g., an elephant). After playing, the study concluded.

# Results

## Reciprocity

All participants in the Prosocial condition chose to share with E2 (100%), compared to 70.8% in the Undermining condition (see Table 1). A Fisher's exact test revealed a significant association between Condition (Undermining vs. Prosocial) and Sharing ($p = .009$). Because no participant chose 'Keep' (no reciprocity) in the Prosocial condition, resulting in a zero cell, the odds ratio could not be reliably estimated. To obtain an interpretable effect size, we conducted a Spearman's rank correlation between condition (0 = Prosocial, 1 = Undermining) and Sharing (0 = no reciprocity, 1 = reciprocity) to estimate effect size. This test revealed a significant association and a medium-to-large effect size ($\rho = -0.41$, $p = .004$). Binomial tests were conducted to assess if reciprocity was above chance within each condition. In the Prosocial condition, children shared significantly above chance ($p < .001$). In the Undermining condition, 71% of participants shared, which was not statistically above chance ($p = .06$). While children consistently reciprocated in the Prosocial condition, undermining behavior reduced the likelihood of doing so.

## Social evaluation

**Liking.** Participants reported strongly liking E2 in both conditions (maximum = 7). In the Prosocial condition, the median liking score was 7 (IQR = 2). In the Undermining condition, the median liking score was also 7 (IQR = 1) (See Table 2). A Wilcoxon test was conducted to compare Liking of E2 between Condition (Undermining vs. Prosocial). The results indicated that there was not a statistically significant difference in Liking of E2 between the two groups ($W = 277.5$, $p = .812$).

**Partner choice.** In the Prosocial condition, 66.7% of participants chose to play with E2 (versus E3), while 45.8% chose to play with E2 in the Undermining condition (See Table 2). A Fisher's exact test was performed to assess the association between Condition (Undermining vs. Prosocial) and Forced Partner Choice (choice between playing with E2 or E3). This test showed no significant association between Condition and Forced Partner Choice ($p = .24$, $OR = 0.43$).

**Social bonding.** Social bonding scores were similar in the Prosocial (median = 3, IQR = 1.5) and Undermining (median = 3, IQR = 2.3) conditions (See Table 2). Ordinal logistic regression was conducted to determine the effect of Condition (Undermining vs. Prosocial) on social bonding. The analysis showed no significant effect of condition ($b = -0.42$, $SE = 0.52$, $z = -0.81$, $p = .42$, 95% CI [−1.45, 0.59]). Condition therefore did not significantly predict the cumulative odds of reporting higher versus lower levels of social bonding.

**Table 1. Number of children who reciprocated by condition.**

| Condition | Keep (no reciprocity) | Give to E2 (reciprocity) |
|---|---|---|
| Prosocial | 0 | 24 |
| Undermining | 7 | 17 |

**Table 2. Descriptive statistics of all dependent variables.**

| Condition | Liking | Social Bonding | Reciprocity | Partner Choice |
|---|---|---|---|---|
| Prosocial | 7 [2] | 3 [1.5] | 100 | 66.7 |
| Undermining | 7 [1] | 3 [2.3] | 70.8 | 45.8 |

Liking and Social Bonding are reported with median and IQR, while Reciprocity and Partner Choice are reported with percentage of participants.

## Discussion

Young children are sensitive to selfish intentions underlying seemingly prosocial behaviors even when they only discover those intentions after the fact. Given this temporal delay, 5-year-old children were significantly less likely to reciprocate towards an agent who revealed a selfish intention compared to an agent who revealed a prosocial intention, holding constant the prosocial action itself. While previous studies demonstrate that children consider intentions revealed before or during prosocial actions, our study uniquely highlights how children respond to an intention revealed after the benefits of a prosocial action. 5-year-olds therefore prioritize prosocial intentions over the benefits of a prosocial action (which have already occurred) when making reciprocity decisions. Children's evaluations of the character, desire for the character to attend a playdate, and desire to wait for the character to return were not significantly different between conditions, suggesting that the selfish intention undermined reciprocity but not broader social evaluations.

From early in life, children's reciprocity behaviors are shaped by intention information [45] and expectations for contingent reciprocity [30]. By age five, these behaviors have become more strategic but also take internalized prosocial norms into account [61], consistent with a two-process moral judgment system emerging between ages four and six [49].

Taken together, our results provide evidence that children considered both the benefit of the prosocial action and the later revealed intention behind the prosocial action. E2's selfish intention appears to have undermined the prosocial action in evaluations of reciprocity in this one-off situation, demonstrating the importance of evaluating intentions. However, the revealed undermining intention did not affect broader social evaluation of E2, indicating that children may rely on an agent's actions or a balance of action and intention to asses an agent's potential as a play partner.

An alternative explanation for our findings is that children's sensitivity to the intention may reflect a sensitivity for equity restoration and not the intention information alone. When E2 revealed that they had received a cookie for letting the child play, children may have inferred that E2 already received a benefit and did not require further reciprocation. Not reciprocating, then, may be viewed as an attempt to make distributions fair or equitable. Children are highly attuned to these inequities from a young age, especially when a distribution is unfavorable to them, though the propensity to act on rectifying these inequities typically arises later than our sample, around six-years-old [64–65]. Further research should be conducted to further parse apart these explanations and may include older children to explore any differences in answers or motivations.

Although children reciprocated significantly less towards the selfish agent, we did not find any significant condition differences in social evaluation. One explanation for this is that children may not have felt that the hidden intention negatively impacted them. Since E2 received a cookie and the child got to play the game, the overall interaction remained positive. Prior research suggests that children are most sensitive to outcomes that cause harm or disadvantage, particularly to themselves [44,66]. The median of liking for E2 in both conditions was at ceiling, potentially suggesting that the prosocial action may have buffered children's evaluations of the agent. However, another explanation for why reciprocity, but not social evaluation, was affected may lie in evidence from belief revision. It is possible that children revised their interpretation of the prosocial act itself, viewing it as less genuine, but without updating their belief about the agent themselves. This revision about the action would affect immediate evaluations, such as reciprocity decisions, but may not affect the longer-term potential for E2 to later be a satisfactory play partner, leaving broader social evaluations unaffected by this one instance. From preschool age, children revise their beliefs when considering counterevidence in their physical and social worlds, so they may be able to revise beliefs in a moral domain as well [67–68]. While our design did not directly measure belief revision, the pattern may be consistent with children updating their moral beliefs regarding the prosocial action. Future work could explore how more negative manipulations may affect social evaluations and test belief revision processes explicitly to understand how children may revise their beliefs in the moral domain.

The prosocial behavior performed by E2, letting the child have a turn in the novel game, was relatively subtle. This subtle prosocial behavior is less overt than other more robust prosocial behaviors, like sharing or helping. It is therefore impressive that even this low-cost act was undermined once a selfish intention was revealed. More explicit prosocial

behaviors may create increased feelings of obligation or expectations of reciprocity, potentially producing even larger reciprocity effects than those observed here. On the other hand, more explicit antisocial behaviors, with either a prosocial or selfish intention, may affect reciprocity and social evaluation measures in the opposite direction. When harm or disadvantage is caused by the agent, this may be strong enough to negatively influence reciprocity decisions and social evaluations [44,66]. However, future research could explore the differences in varying prosocial acts.

One limitation of the study is that the sample size was small and only included one age group: 5-year-olds. Still, our effect size was moderate-to-large even given the small sample, but future research with larger samples should be conducted to replicate the findings presented here. Future research may also include a wider age range. Given that children younger than five often focus on outcomes instead of intentions and that theory of mind reasoning is still developing through age four [69–71] we would expect that 3- to 4-year-olds would show a weaker or inconsistent sensitivity to E2's selfish intention. On the other hand, we would expect older children to rely more on intentions and norm reasoning [3] that would lead to a similar, though potentially stronger sensitivity to E2's selfish intention. We may also expect to find more cross-cultural differences as norms may vary across cultures and prosocial behaviors change as a function of culture around age six [62]. Future research should expand this paradigm to a larger age range to understand key developmental differences and should examine any differences cross-culturally, especially as children's prosocial norms become more codified in middle childhood.

Another limitation of our study is that it was carried out during the COVID-19 pandemic, so it was conducted entirely over Zoom. However, recent research suggests that conducting studies online versus in person show few to no differences and may even increase diversity in samples [72–74]. Still, perhaps the selfish intention and the outcome would have been more salient to children had the cookie been a material good that the child could see. Future studies could adapt our paradigm for an in-person context to see if social evaluations of the agent may be more negatively impacted by this in-person method.

Overall, we provide evidence that children engage in strategic reciprocity when given intention information revealing a selfish intention after receiving the benefit of a prosocial action. While social evaluation did not differ by condition, our results follow theories of early emerging contingent reciprocity in children. The present study illustrates that selfish intentions revealed after a prosocial action negatively impact reciprocity decisions in young children.

## Supporting information

**S1 File. Study datasheet.** All data are reported.
(XLSX)

**S2 File. Procedure script.** Exact script used for all sessions.
(DOCX)

## Acknowledgments

We thank Bella Larsen for helping with stimuli creation. We also thank the Tomasello Lab research assistants who helped recruit and code for this project.

## Author contributions

**Conceptualization:** Kayley Dotson, Michael Tomasello.

**Investigation:** Kayley Dotson.

**Methodology:** Kayley Dotson.

**Resources:** Michael Tomasello.

**Supervision:** Michael Tomasello.

**Visualization:** Kayley Dotson.

**Writing – original draft:** Kayley Dotson.

**Writing – review & editing:** Michael Tomasello.

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
