## [Decision Letter · Decision Letter 0]

4 Oct 2025

Dear Dr. Dotson,

Thank you for submitting your manuscript to PLOS ONE. After careful consideration, we feel that it has merit but does not fully meet PLOS ONE’s publication criteria as it currently stands. Therefore, we invite you to submit a revised version of the manuscript that addresses the points raised during the review process.

We look forward to receiving your revised manuscript.

Kind regards,

Asami Shinohara

Academic Editor

PLOS ONE

**Journal Requirements:**

1. When submitting your revision, we need you to address these additional requirements. Please ensure that your manuscript meets PLOS ONE's style requirements, including those for file naming. The PLOS ONE style templates can be found at https://journals.plos.org/plosone/s/file?id=wjVg/PLOSOne_formatting_sample_main_body.pdf and https://journals.plos.org/plosone/s/file?id=ba62/PLOSOne_formatting_sample_title_authors_affiliations.pdf 2. Your ethics statement should only appear in the Methods section of your manuscript. If your ethics statement is written in any section besides the Methods, please move it to the Methods section and delete it from any other section. Please ensure that your ethics statement is included in your manuscript, as the ethics statement entered into the online submission form will not be published alongside your manuscript. 3. We note that there is identifying data in the Supporting Information file. Due to the inclusion of these potentially identifying data, we have removed this file from your file inventory. Prior to sharing human research participant data, authors should consult with an ethics committee to ensure data are shared in accordance with participant consent and all applicable local laws. Data sharing should never compromise participant privacy. It is therefore not appropriate to publicly share personally identifiable data on human research participants. The following are examples of data that should not be shared: -Name, initials, physical address-Ages more specific than whole numbers-Internet protocol (IP) address-Specific dates (birth dates, death dates, examination dates, etc.)-Contact information such as phone number or email address-Location data-ID numbers that seem specific (long numbers, include initials, titled “Hospital ID”) rather than random (small numbers in numerical order) Data that are not directly identifying may also be inappropriate to share, as in combination they can become identifying. For example, data collected from a small group of participants, vulnerable populations, or private groups should not be shared if they involve indirect identifiers (such as sex, ethnicity, location, etc.) that may risk the identification of study participants. Additional guidance on preparing raw data for publication can be found in our Data Policy (https://journals.plos.org/plosone/s/data-availability#loc-human-research-participant-data-and-other-sensitive-data) and in the following article: http://www.bmj.com/content/340/bmj.c181.long. Please remove or anonymize all personal information (<specific identifying information in file to be removed>), ensure that the data shared are in accordance with participant consent, and re-upload a fully anonymized data set. Please note that spreadsheet columns with personal information must be removed and not hidden as all hidden columns will appear in the published file. 4. Please include captions for your Supporting Information files at the end of your manuscript, and update any in-text citations to match accordingly. Please see our Supporting Information guidelines for more information: http://journals.plos.org/plosone/s/supporting-information. 5. If the reviewer comments include a recommendation to cite specific previously published works, please review and evaluate these publications to determine whether they are relevant and should be cited. There is no requirement to cite these works unless the editor has indicated otherwise. 

**Additional Editor Comments:**

Thank you for submitting your manuscript to PLOS ONE and for your patience during the review process.

Your manuscript has been evaluated by peer reviewers with expertise in your field.

While the reviewers acknowledged the strengths of your work, they also raised significant concerns that must be addressed before it can be considered for publication.

In particular, the reviewers require you to clarify how the sample size was determined. They also noted errors in the statistical analysis, which should be carefully addressed.

Please review the reviewers’ comments and revise your manuscript with point-by-point responses. We do not encourage multiple rounds of major revision. Therefore, if the next revision does not provide satisfactory responses to the reviewers’ concerns, we will have to consider rejecting the manuscript.

We look forward to receiving your revised submission.

Reviewers' comments:

**Comments to the Author**

1. Is the manuscript technically sound, and do the data support the conclusions?

Reviewer #1: Yes

Reviewer #2: No

Reviewer #3: Yes

Reviewer #4: Partly

2. Has the statistical analysis been performed appropriately and rigorously?

Reviewer #1: Yes

Reviewer #2: No

Reviewer #3: Yes

Reviewer #4: No

3. Have the authors made all data underlying the findings in their manuscript fully available?

Reviewer #1: No

Reviewer #2: Yes

Reviewer #3: Yes

Reviewer #4: Yes

4. Is the manuscript presented in an intelligible fashion and written in standard English?

Reviewer #1: Yes

Reviewer #2: Yes

Reviewer #3: Yes

Reviewer #4: Yes

**Reviewer #1:**  This paper examines how children reason about prosocial actions when a selfish intention is revealed after a prosocial behavior. The main finding is that when given a chance to reciprocate a prosocial act, children shared less when the puppet had previously demonstrated an undermining intention compared to a neutral one. However, children in the undermining condition did not show reduced liking or social bonding toward the selfish agent. I believe this paper makes a meaningful contribution to the field of prosocial development. I have several suggestions and questions regarding the current version of the manuscript:

1. On page 5, line 84, the authors state that they are interested in what happens when children initially believe one thing about a partner’s intentions and then receive contradictory evidence. They also mention they aim to investigate whether children engage in moral belief revision. However, the methodology does not include a measure of children’s beliefs about the partner’s initial intentions, nor does it directly assess belief revision. It is therefore unclear whether children inferred any intention at all prior to its revelation, or whether they had already assumed a selfish motive from the beginning.

2. I suggest the authors provide more justification for focusing solely on 5-year-olds, rather than including a broader age range. How might developmental differences affect the findings?

3. How was the sample size determined? Is it possible that the study is underpowered?

4. I found the procedure somewhat difficult to follow. In particular, the role of E3 was not immediately clear. I recommend providing a clearer description of E3’s function in the procedure.

5. The prosocial act performed by E2—allowing the child to play first—is relatively subtle. Would the results differ if the prosocial behavior were more direct or explicit, such as sharing or helping?

6. In the results section, I suggest including descriptive statistics for the liking, forced-choice, and social bonding measures. Additionally, it would be helpful to conduct analyses comparing performance to chance for each condition. For example, did children reciprocate significantly above chance in both the neutral and undermining conditions?

**Reviewer #2:** Summary

This study examined how the revelation of selfish intentions behind prosocial behavior affects children's reciprocity. When a puppet's prosocial behavior was followed by the revelation of selfish intent, 5-year-olds were less likely to behave reciprocally toward the puppet compared to when neutral (prosocial) intent was revealed. However, no differences were found between conditions in children's liking of the agent or social bonding measures.

General Comment

This paper addresses an interesting question about how children use information about hidden selfish intentions behind prosocial behavior in their reciprocal actions. Indeed, this research could potentially contribute new insights to our understanding of the development of reciprocity in children. However, this manuscript does not yet sufficiently meet the criteria upheld by this journal, particularly regarding research framing, erroneous analyses, and insufficient discussion.

Major Concerns

1. Research Framing

The introduction and discussion appear to be structured around two key concepts: reciprocity and belief revision. While I can understand that this is a study about reciprocity, the study's procedures and results do not support the claim that this is a study of belief revision.

The study's procedures only measure children's behavior and evaluations of/choices about the agent after intention disclosure. Therefore, the study fails to establish: (1) what beliefs children initially held about the prosocial agent, and (2) whether these beliefs changed "before and after" the presentation of intention information. The relationship between beliefs about prosocial behavior and evaluations of the agent (or whether such evaluations are included in beliefs) also remains unclear.

Therefore, what can be concluded from this study's results is simply that children's reciprocal behavior decreases when selfish intentions behind prosocial behavior are revealed. Consequently, the introduction and discussion need to be restructured to focus specifically on children's reciprocity and the relationship between prosocial behavior and intention.

2. Critical Errors in Statistical Analysis

The reporting of analytical results in this paper contains several critical errors and insufficiencies.

Insufficient Data Reporting: For the Liking score, Forced Choice, and Social Bonding results, only analytical results (mainly p-values) are described, making it completely unclear what the actual data patterns were. For example, regarding the Liking score, while we understand there was no group difference, it's unclear whether this means the agent was liked in both groups (mean scores close to 7) or disliked (close to 1). Without these detailed results, readers struggle to understand the findings, and any discussion derived from the data becomes thin or erroneous.

Social Bonding Analysis Errors: More problematic is the analysis and interpretation of Social Bonding results:

1. Incorrect test statistic: The authors report "t = -0.814" for ordinal logistic regression, but ordinal logistic regression uses Wald z-statistics or likelihood ratio tests, never t-statistics. The correct analysis and statistical values need verification.

2. Inappropriate interpretation of non-significant results: Despite the model being statistically non-significant (evident from both the p-value and 95% CI crossing zero), the authors inappropriately interpret odds ratios. When results are non-significant, such effect size interpretations should be avoided.

3. Sign inconsistency: The authors state log odds = -0.42, but then write "exp(.42) = 1.52 times higher," with the sign reversed. Following the log odds value, it should be "1.52 times lower."

4. Misinterpretation of ordinal logistic regression: The odds scale interpretation treats this as a binary comparison ("not waiting at all vs waiting after one or more prompts"). However, the coefficient represents the change in cumulative log odds for being in lower versus higher categories across all possible thresholds, not just a single binary comparison.

3. Insufficient Discussion

The discussion is superficially shallow overall, with little reference to prior research, resulting in mostly simple result descriptions and mere speculation. This is particularly problematic given that belief revision cannot be addressed. For example, regarding why group differences were observed in reciprocal behavior but not in liking or social bonding, many points remain to be discussed: why such results occurred, whether both results would be expected to align in older children or adults, and if so, what developmental factors might be involved. Even Vaish et al. (2018), cited in this paper, found that behavioral and evaluative results did not necessarily align, but they provided sufficient discussion of this point. Without deepening the discussion by referencing such prior research, this paper remains merely a report of results, with limited scholarly contribution.

Other Concerns & Minor comments

Unclear sample size determination:

While the study included 48 participants, the rationale for determining this sample size is not provided. Although 48 participants may not be considered too small, demonstrating that the sample size and analyses are appropriate is essential for establishing this as rigorous scientific psychological research.

Disorganized literature review:

Throughout the introduction, the positioning of prior research findings and this study is poorly organized, giving the impression of a list of prior studies related to keywords like prosocial behavior, moral judgment, intention, and reciprocity. What's important for deriving this study's research question is: (1) children engage in prosocial behavior and evaluate prosocial people positively, (2) reciprocity toward others' prosocial behavior is observed in children, and (3) intention information is used in evaluations of others and reciprocity. The literature review should focus on these points, then explain how this study's questions differ from prior research.

Unclear emphasis on "after":

Several places emphasize "after" in "selfish intention revealed after a prosocial behavior" with italics, but it's unclear why "after" is considered so important. This study did not examine the effects of when selfish intentions are revealed, nor was this timing emphasized in discussions of prior research. While this study does have a structure where selfish intentions are revealed after prosocial behavior, emphasizing "after" may mislead readers about the main research objective (i.e., whether selfish intent affects reciprocity).

Missing exclusion criteria:

The raw data suggests that several participants were excluded from analysis, but there is no description of exclusion criteria or the number of excluded participants. If specific exclusion criteria were established, they need to be clearly stated.

**Reviewer #3:** The manuscript examines how five‑year‑old children update social decisions when a helper’s selfish motive is revealed after a prosocial act. Primary result: reciprocity dropped when the selfish intention was revealed.

I think this paper has the potential to make a valuable contribution to the literature and can be published after a moderate revision.

Comments

1) The “neutral” condition is not neutral. The control puppet states, “I only let you play because I wanted you to have fun,” which is explicitly other‑regardingand morally positive, not neutral. This makes the contrast “self‑regarding vs. other‑regarding,” not “undermining vs. neutral,” and it may inflate differences in reciprocity by juxtaposing a plainly benevolent motive against a plainly selfish one. I recommend renaming conditions accordingly and, ideally, adding a true neutral motive.

2) Alternative explanation: equity repair rather than “moral belief revisionn. When E2 announces they got a cookie for letting the child play, children may infer an advantage accrued to E2. Choosing not to give the additional toy to E2 can be read as equity restoration (a fairness decision), not necessarily a revision of moral beliefs about intentions per se. The present design cannot distinguish “motive‑based discounting” from “resource‑balancing.”

3) Primary effect size and its reporting. The paper reports p = .009, but no effect size. Please report also an effect size.

4) Statistical inconsistency in the social‑bonding analysis. In the Results you write that the ordinal‑logit coefficient for the undermining condition is −0.42 (95% CI [−1.45, 0.59]) but then interpret the odds as exp(.42) = 1.52 times higher—this reverses the sign of the coefficient and converts a non‑significant, negative effect into a positive one. I think the correct transformation of −0.42 is exp(−0.42) ≈ 0.66 (i.e., ~34% lower odds), and the effect is non‑significant and should be interpreted accordingly.

5) Methodological details. You report a final N = 48 (24/condition) but do not state how many children were initially recruited, how many were excluded (e.g., those who did not say “now”), or why. CONSORT‑style accounting (or a simple flow diagram) is needed to assess selection bias. Please also specify how condition was randomly assigned, whether assignment was balanced across time of day/experimenter, and whether the live experimenter (E1) was blind to condition during outcome collection. Pre‑registration (if any) should be declared; otherwise, please say so explicitly.

6) Theoretical positioning. There is quite a lot of literature on moral preferences among adults, which I think is relevant here (e.g., Alger & Weibull, 2013; Capraro & Rand, 2018; Capraro & Perc, 2021; Basic & Verrina, 2024).

References

Alger, I., & Weibull, J. W. (2013). Homo moralis—preference evolution under incomplete information and assortative matching. Econometrica, 81(6), 2269-2302.

Bašić, Z., & Verrina, E. (2024). Personal norms—and not only social norms—shape economic behavior. Journal of Public Economics, 239, 105255.

Capraro, V., & Rand, D. G. (2018). Do the right thing: Experimental evidence that preferences for moral behavior, rather than equity or efficiency per se, drive human prosociality. Judgment and Decision Making, 13(1), 99-111.

Capraro, V., & Perc, M. (2021). Mathematical foundations of moral preferences. Journal of the Royal Society interface, 18(175), 20200880.

**Reviewer #4:** This manuscript investigates how preschool-aged children evaluate prosocial actions when selfish intentions are revealed later on. Using a Zoom-based puppet paradigm, the authors tested whether five-year-old children would reciprocate prosocial behavior differently depending on whether the prosocial act was later framed as motivated by selfish versus neutral intentions. The study itself is well-designed and on an interesting topic within the domain of prosociality and intentions. However, I have a number of concerns about the manuscript:

My biggest concern is the review of the literature, both in breadth and recency. The reference list is extremely brief for such a well-studied topic, with one-fourth coming from the same research group and no citations beyond the year 2020. Several arguments made seem to be lacking appropriate citations. For example, the sections on children’s sensitivity to intentions can include works by Baird, Astington, Killen, Carey, Nielsen, Blake, and many more. Research by Baillargeon, Smetana, Harris, and others come to mind on the topic of the early development of reciprocity. The authors might also consider discussing connections to moral reasoning frameworks (e.g., Killen, Rutland). The literature included is also heavily US-centric, when many of the findings discussed have been replicated in other countries, which significantly undermines the impact of the work. On the topic of contradictory evidence, work on children’s ability to consider counterfactuals may help inform hypotheses and/or make sense of the results. Overall, the research groups I’ve suggested is only a small portion of the relevant literature.

The methodology and results section were very brief and skewed informal in tone. Despite a very small sample size, there is no discussion of power analysis to support or justify it; as a result, I am skeptical about the rigorousness of the statistical analyses conducted. There is no discussion of preregistration -- and if that is the case this should be acknowledged in the main text. Statistical reporting do not follow APA style. Description of the measures are inconsistent (for example, for social evaluation the authors specify it is forced choice, but do not elaborate for the reciprocity measure).

For discussion, the exploration of why they find a significant difference in reciprocity but not social evaluation is very brief and could be expanded – likely after a more thorough literature review is conducted on the topic. There is also no explicit discussion for the social bonding results. Overall, the contribution is difficult to judge due to the lack of depth in interpreting the results.

**Do you want your identity to be public for this peer review?** For information about this choice, including consent withdrawal, please see our Privacy Policy

Reviewer #1: No

Reviewer #2: **Yes:** Rizu Toda

Reviewer #3: No

Reviewer #4: No

---

## [Author Response · Author response to Decision Letter 1]

18 Nov 2025

Thank you for the opportunity to submit a revision of our manuscript entitled “When revealed after the fact, selfish intentions undermine prosocial actions in five-year-olds”. We appreciate the thoughtful and constructive feedback that you and the reviewers provided. We have revised the manuscript to properly match PLOS ONE’s style requirements, moved our ethics statement to the proper location, removed all identifiers from our data, and added captions for Supporting Information. We have worked to address each comment and believe the manuscript is far stronger now. Below, we provide a point-by-point response for each reviewer comment.

Reviewer 1

1. On page 5, line 84, the authors state that they are interested in what happens when children initially believe one thing about a partner’s intentions and then receive contradictory evidence. They also mention they aim to investigate whether children engage in moral belief revision. However, the methodology does not include a measure of children’s beliefs about the partner’s initial intentions, nor does it directly assess belief revision. It is therefore unclear whether children inferred any intention at all prior to its revelation, or whether they had already assumed a selfish motive from the beginning.

Response: Thank you for your thoughtful feedback. We agree that our original framing overstated what our design measured and have now removed any claims that our study explores belief revision. Now, our only mention of belief revision comes in the discussion as a point for future research:

“Another explanation for why reciprocity, but not social evaluation or social bonding, was affected may lie in evidence from belief revision. It is possible that children revised their interpretation of the prosocial act itself, viewing it as less genuine, but without updating their belief about the agent themselves. Preschool children revise their beliefs when considering counterevidence in their physical and social worlds (Kimura & Gopnik, 2019; Migosa et al., 2020). While our design did not directly measure belief revision, the pattern is consistent with children updating their moral beliefs regarding the prosocial action. Future work should test these processes explicitly to understand how children may revise their beliefs in the moral domain.”

2. I suggest the authors provide more justification for focusing solely on 5-year-olds, rather than including a broader age range. How might developmental differences affect the findings?

Response: Thank you for raising this point. We now explicitly justify our focus on 5-year-olds in the introduction. We also added a section in the discussion to mention the need for future work to explore a larger age range. In the introduction we have added the following justification:

“Exploring how 5-year-olds reason about prosocial motivations may be of particular interest due to recent theoretical accounts and empirical findings. Recent theoretical models propose two major shifts in prosocial development: from early intrinsic motivation to strategic concerns by age five (Engelmann & Rapp, 2018; Leimgruber, 2018; Robbins & Rochat, 2011; Warneken et al., 2019) and then from strategic concerns to internalized prosocial and fairness norms around and beyond age five (Blake et al., 2015; House, 2018; Hepach and Tomasello, 2025). Also, children’s prosocial behaviors at this age may be more stable across culture. Across six societies, prosocial behaviors did not change as a function of culture until around age six (House et al., 2013). Five-year-old children therefore are at a transitional point, wherein they can understand and strategically engage in prosocial behaviors and reciprocity but are not yet necessarily fully guided by moral or cultural norms. This age therefore provides an ideal window into investigating how intention information shapes prosocial behavior and reciprocity.”

In the discussion, we have added possible developmental differences and the need to explore these age ranges further: “One limitation of the study is that the sample size was small and only included one age group: 5-year-olds. Still, our effect size was moderate-to-large even given the small sample, but future research with larger samples should be conducted to replicate the findings presented here. Future research may also include a wider age range. Given that children younger than five often focus on outcomes instead of intentions and that theory of mind reasoning is still developing through age four (Astington & Jenkins, 1995; Baird & Astington, 2004; Wellman et al., 1990), we would expect that 3- to 4-year-olds would show a weaker or inconsistent sensitivity to E2’s selfish intention. On the other hand, we would expect older children to rely more on intentions and norm reasoning (Killen & Smetana, 2013) that would lead to a similar, though potentially stronger sensitivity to E2’s selfish intention.”

3. How was the sample size determined? Is it possible that the study is underpowered?

Response: We have included the following justification for our sample size in our methods section: “A sample size of 48 was selected as it is conventional in the field and, in this case, slightly larger given that the study focused on a single age group (see Beeler-Duden & Vaish, 2020; Vasil et al., 2024).”

4. I found the procedure somewhat difficult to follow. In particular, the role of E3 was not immediately clear. I recommend providing a clearer description of E3’s function in the procedure.

Response: Thank you for pointing this out. We have added the following additional information about E3’s role in the procedure: “E3 was introduced at the beginning to serve as a neutral, somewhat positive agent. E3 would later be included as a potential play partner in our dependent measures.”

5. The prosocial act performed by E2—allowing the child to play first—is relatively subtle. Would the results differ if the prosocial behavior were more direct or explicit, such as sharing or helping?

Response: We agree that the prosocial manipulation is rather subtle. We have added a paragraph in our discussion to expand on this idea and highlight the need for future research on this:

“The prosocial behavior performed by E2, letting the child have a turn in the novel game, was relatively subtle. This subtle prosocial behavior is less overt than other more robust prosocial behaviors, like sharing or helping. It is therefore impressive that even this low-cost act was undermined once a selfish intention was revealed. More explicit prosocial behaviors may create increased feelings of obligation or expectations of reciprocity, potentially producing even larger reciprocity effects than those observed here. However, future research could explore the differences in varying prosocial acts.”

6. In the results section, I suggest including descriptive statistics for the liking, forced-choice, and social bonding measures. Additionally, it would be helpful to conduct analyses comparing performance to chance for each condition. For example, did children reciprocate significantly above chance in both the neutral and undermining conditions?

Response: Thank you for suggesting this. We have now added a table of descriptive statistics for all DVs, included these descriptives throughout our results section, and conducted and included analyses comparing reciprocity to chance.

Reviewer 2

Major Concerns

1. Research Framing

The introduction and discussion appear to be structured around two key concepts: reciprocity and belief revision. While I can understand that this is a study about reciprocity, the study's procedures and results do not support the claim that this is a study of belief revision.

The study's procedures only measure children's behavior and evaluations of/choices about the agent after intention disclosure. Therefore, the study fails to establish: (1) what beliefs children initially held about the prosocial agent, and (2) whether these beliefs changed "before and after" the presentation of intention information. The relationship between beliefs about prosocial behavior and evaluations of the agent (or whether such evaluations are included in beliefs) also remains unclear.

Therefore, what can be concluded from this study's results is simply that children's reciprocal behavior decreases when selfish intentions behind prosocial behavior are revealed. Consequently, the introduction and discussion need to be restructured to focus specifically on children's reciprocity and the relationship between prosocial behavior and intention.

Response: Thank you for pointing this out. We have now removed any claims that our study explores belief revision. We also have reframed the study to highlight prosocial preferences, reciprocity, and intentions and have arranged our introduction accordingly. Now, our only mention of belief revision comes in the discussion as a point for future research:

“Another explanation for why reciprocity, but not social evaluation or social bonding, was affected may lie in evidence from belief revision. It is possible that children revised their interpretation of the prosocial act itself, viewing it as less genuine, but without updating their belief about the agent themselves. Preschool children revise their beliefs when considering counterevidence in their physical and social worlds (Kimura & Gopnik, 2019; Migosa et al., 2020). While our design did not directly measure belief revision, the pattern is consistent with children updating their moral beliefs regarding the prosocial action. Future work should test these processes explicitly to understand how children may revise their beliefs in the moral domain.”

2. Critical Errors in Statistical Analysis

The reporting of analytical results in this paper contains several critical errors and insufficiencies.

Insufficient Data Reporting: For the Liking score, Forced Choice, and Social Bonding results, only analytical results (mainly p-values) are described, making it completely unclear what the actual data patterns were. For example, regarding the Liking score, while we understand there was no group difference, it's unclear whether this means the agent was liked in both groups (mean scores close to 7) or disliked (close to 1). Without these detailed results, readers struggle to understand the findings, and any discussion derived from the data becomes thin or erroneous.

Response: Thank you for suggesting this. We have now added a table of descriptive statistics for all DVs, included these descriptives throughout our results section, and conducted and included analyses comparing reciprocity to chance.

Social Bonding Analysis Errors: More problematic is the analysis and interpretation of Social Bonding results:

1. Incorrect test statistic: The authors report "t = -0.814" for ordinal logistic regression, but ordinal logistic regression uses Wald z-statistics or likelihood ratio tests, never t-statistics. The correct analysis and statistical values need verification.

Response: Thank you for highlighting this error. We have fixed this in the results section to show the accurate Wald z-statistic.

2. Inappropriate interpretation of non-significant results: Despite the model being statistically non-significant (evident from both the p-value and 95% CI crossing zero), the authors inappropriately interpret odds ratios. When results are non-significant, such effect size interpretations should be avoided.

Response: Thank you for highlighting this. We have removed any interpretations of the non-significant results.

3. Sign inconsistency: The authors state log odds = -0.42, but then write "exp(.42) = 1.52 times higher," with the sign reversed. Following the log odds value, it should be "1.52 times lower."

Response: We have removed this entirely, so as not to make any erroneous claims.

4. Misinterpretation of ordinal logistic regression: The odds scale interpretation treats this as a binary comparison ("not waiting at all vs waiting after one or more prompts"). However, the coefficient represents the change in cumulative log odds for being in lower versus higher categories across all possible thresholds, not just a single binary comparison.

Response: Again, we have removed this entirely, so as not to make any erroneous claims.

3. Insufficient Discussion

The discussion is superficially shallow overall, with little reference to prior research, resulting in mostly simple result descriptions and mere speculation. This is particularly problematic given that belief revision cannot be addressed. For example, regarding why group differences were observed in reciprocal behavior but not in liking or social bonding, many points remain to be discussed: why such results occurred, whether both results would be expected to align in older children or adults, and if so, what developmental factors might be involved. Even Vaish et al. (2018), cited in this paper, found that behavioral and evaluative results did not necessarily align, but they provided sufficient discussion of this point. Without deepening the discussion by referencing such prior research, this paper remains merely a report of results, with limited scholarly contribution.

Response: We appreciate your feedback on our discussion and have strengthened this section by adding a few new paragraphs and several citations to support our ideas. We now include authors like Blake, Shaw, Killen, Astington, Baird, Wellman, and House among others to discuss alternative explanations, potential age-related differences, and theoretical positioning.

Other Concerns & Minor comments

Unclear sample size determination:

While the study included 48 participants, the rationale for determining this sample size is not provided. Although 48 participants may not be considered too small, demonstrating that the sample size and analyses are appropriate is essential for establishing this as rigorous scientific psychological research.

Response: We have included the following justification for our sample size in our methods section: “A sample size of 48 was selected as it is conventional in the field and, in this case, slightly larger given that the study focused on a single age group (see Beeler-Duden & Vaish, 2020; Vasil et al., 2024).”

Disorganized literature review:

Throughout the introduction, the positioning of prior research findings and this study is poorly organized, giving the impression of a list of prior studies related to keywords like prosocial behavior, moral judgment, intention, and reciprocity. What's important for deriving this study's research question is: (1) children engage in prosocial behavior and evaluate prosocial people positively, (2) reciprocity toward others' prosocial behavior is observed in children, and (3) intention information is used in evaluations of others and reciprocity. The literature review should focus on these points, then explain how this study's questions differ from prior research.

Response: Thank you again for your thoughtful reading of the introduction. We have reorganized the introduction to largely follow the suggestions you laid out. We introduce literature on prosocial behaviors, reciprocity, and intentions and frame our study with these literatures.

Unclear emphasis on "after":

Several places emphasize "after" in "selfish intention revealed after a prosocial behavior" with italics, but it's unclear why "after" is considered so important. This study did not examine the effects of when selfish intentions are revealed, nor was this timing emphasized in discussions of prior research. While this study does have a structure where selfish intentions are revealed after prosocial behavior, emphasizing "after" may mislead readers about the main research objective (i.e., whether selfish intent affects reciprocity).

Response: As mentioned above, we have now removed any claims that our study explores belief revision. We also have reframed the study to highlight prosocial preferences, reciprocity, and intentions and have arranged our introduction accordingly.

Missing exclusion criteria:

The raw data suggests that several participants were excluded from analysis, but there is no description of exclusion criteria or the number of excluded participants. I

---

## [Decision Letter · Decision Letter 1]

16 Dec 2025

Dear Dr. Dotson,

Thank you for submitting your manuscript to PLOS ONE. After careful consideration, we feel that it has merit but does not fully meet PLOS ONE’s publication criteria as it currently stands. Therefore, we invite you to submit a revised version of the manuscript that addresses the points raised during the review process.

We look forward to receiving your revised manuscript.

Kind regards,

Asami Shinohara

Academic Editor

PLOS One

Journal Requirements:

Additional Editor Comments:

Your manuscript has been evaluated by the peer reviewers from the previous round.

While two reviewers accepted the paper, one reviewer still has concerns that must be addressed before it can be considered for publication.

Therefore, I invite you to revise your manuscript.

Please review the reviewer’s comments and revise your manuscript with point-by-point responses.

We look forward to receiving your revised submission.

Reviewers' comments:

Reviewer's Responses to Questions

**Comments to the Author**

Reviewer #1: All comments have been addressed

Reviewer #2: (No Response)

Reviewer #3: All comments have been addressed

2. Is the manuscript technically sound, and do the data support the conclusions?

Reviewer #1: Yes

Reviewer #2: Partly

Reviewer #3: Yes

3. Has the statistical analysis been performed appropriately and rigorously?

Reviewer #1: Yes

Reviewer #2: Yes

Reviewer #3: Yes

4. Have the authors made all data underlying the findings in their manuscript fully available?

Reviewer #1: Yes

Reviewer #2: Yes

Reviewer #3: Yes

5. Is the manuscript presented in an intelligible fashion and written in standard English?

Reviewer #1: Yes

Reviewer #2: Yes

Reviewer #3: Yes

Reviewer #1: I find the revisions satisfactory and do not have further comments. The authors have addressed all of my concerns raised in the previous round of review with care and clarity.

Reviewer #2: Thank you for the opportunity to review this research again. I acknowledge that the authors have made numerous revisions in response to reviewer feedback, particularly regarding the framing of the study and the analysis and interpretation of results. However, these revisions leave fundamental issues concerning the positioning and significance of this research unresolved, and raise additional concerns.

The primary issue is that the shallow discussion in the Introduction and Discussion remains unaddressed. While the authors' decision to remove the brief revision discussion and focus on the relationship between reciprocity and intention has improved alignment between the Introduction and the experimental content, the literature review remains superficial. As Reviewers 3 and 4 noted in the previous round, prosocial behavior and reciprocity are extensively researched topics. Due to the insufficient review and discussion of this substantial body of work, it remains unclear what novel contribution this study makes to this well-established field.

More specifically, the Introduction notes that numerous studies have already demonstrated that young children consider agents' intentions when engaging in prosocial and reciprocal behaviors. What, then, distinguishes this study from prior work? Why does the current research question arise from these existing findings? The current Introduction presents prior findings and the current study in somewhat disconnected manner, obscuring the study's significance.

Below are additional detailed comments on specific sections.

<introduction>

In reviewing the literature on prosociality and reciprocity, findings from infants, toddlers, and preschoolers are sometimes conflated under the term "young children." Given that age is crucial for understanding development, the review should clearly specify which developmental stage each finding pertains to.

The Introduction presents research on adult behavioral economics and introduces the homo moralis model, which appears irrelevant to this study. If this model underlies the research and testing its applicability to children is important, this should be discussed in the Discussion section (currently, it is mentioned only in this Introduction paragraph). Otherwise, it is misleading and should either be removed or reframed.

The rationale for including "willingness to play with E2" and "social bonding" as dependent variables is unclear. I understood that prize allocation to E2 measures reciprocity and liking scores assess evaluation of E2. The purpose of the remaining two measures needs explanation in either the Introduction or Methods. Clarifying this would enable deeper discussion of why reciprocity showed condition differences while the other three measures did not.

<methods>

The Methods section remains unclear in places. For example, were questions about reciprocity, social evaluation, and social bonding counterbalanced? How was the new toy (colorful bouncy ball) introduced to children?

<results>

P-value reporting is inconsistent (e.g., p = 0.009, p < 0.01, p = .812). Please standardize the format for accurate reporting.

<discussion>

Lines 279-281 (p. 14) state: "Taken together, our results provide evidence that children considered the prosocial action, the revealed intention, and their own personal benefits." However, the experiment does not demonstrate consideration of personal benefits. Since the ball given to E2 belonged to E1, not the participant, it had no impact on the participant's personal benefits.

Lines 300-307 (p. 15) propose belief revision to explain why only reciprocity was affected by intention. However, this explanation is insufficient for why reciprocity alone was influenced. Moreover, given the partial effects observed, it is premature to conclude that children revise moral beliefs similarly to physical and social beliefs. If discussing belief revision, you must explain why intention information revised beliefs about the prosocial act but left beliefs about the agent unaffected.

In summary, while the authors' substantial revisions have improved the manuscript from its initial version, it remains unclear what contribution this research makes to the extensive existing literature on this topic.</discussion></results></methods></introduction>

Reviewer #3: (No Response)

**Do you want your identity to be public for this peer review?** For information about this choice, including consent withdrawal, please see our Privacy Policy

Reviewer #1: No

Reviewer #2: No

Reviewer #3: No

---

## [Author Response · Author response to Decision Letter 2]

27 Jan 2026

Thank you for the opportunity to submit revisions for our manuscript entitled “When revealed after the fact, selfish intentions undermine prosocial actions in five-year-olds”. We appreciate the careful feedback provided to us for this round of revisions. Reviewers 1 and 3 were satisfied with our revisions, but Reviewer 2 still had some concerns. We have addressed each comment provided by this reviewer. Below, we provide a point-by-point response for each comment.

Reviewer 2

1. Thank you for the opportunity to review this research again. I acknowledge that the authors have made numerous revisions in response to reviewer feedback, particularly regarding the framing of the study and the analysis and interpretation of results. However, these revisions leave fundamental issues concerning the positioning and significance of this research unresolved, and raise additional concerns.

The primary issue is that the shallow discussion in the Introduction and Discussion remains unaddressed. While the authors' decision to remove the brief revision discussion and focus on the relationship between reciprocity and intention has improved alignment between the Introduction and the experimental content, the literature review remains superficial. As Reviewers 3 and 4 noted in the previous round, prosocial behavior and reciprocity are extensively researched topics. Due to the insufficient review and discussion of this substantial body of work, it remains unclear what novel contribution this study makes to this well-established field.

More specifically, the Introduction notes that numerous studies have already demonstrated that young children consider agents' intentions when engaging in prosocial and reciprocal behaviors. What, then, distinguishes this study from prior work? Why does the current research question arise from these existing findings? The current Introduction presents prior findings and the current study in somewhat disconnected manner, obscuring the study's significance.

Response: Thank you for this feedback. We have worked to improve the flow, clarity, and depth of our introduction and discussion. We agree that the study's significance was previously muddied by the organization. We have now improved the flow of both our introduction and discussion and frame the significance of our study more explicitly. The key point is perhaps expressed most succinctly on p. 6: “We therefore ask how children reason about an agent when a self-interested motivation for a seemingly prosocial action is revealed only after the fact. Since the prosocial action has already taken place and children will have already received its benefit, children must decide how to balance the prosocial action itself and the intention underlying the action in their reciprocity decisions.”

2. In reviewing the literature on prosociality and reciprocity, findings from infants, toddlers, and preschoolers are sometimes conflated under the term "young children." Given that age is crucial for understanding development, the review should clearly specify which developmental stage each finding pertains to.

Response: Thank you for pointing this out. We have added specific ages for findings in both our introduction and discussion to clarify these ages and strengthen our argument.

3. The Introduction presents research on adult behavioral economics and introduces the homo moralis model, which appears irrelevant to this study. If this model underlies the research and testing its applicability to children is important, this should be discussed in the Discussion section (currently, it is mentioned only in this Introduction paragraph). Otherwise, it is misleading and should either be removed or reframed.

Response: We agree that the way the homo moralis model as presented seemed disconnected at best. We have incorporated the model into the introduction as a smaller point that reinforces these ideas as connected to children. It now reads as follows:

“Reciprocity decisions and social evaluations therefore require children to balance sources of information, including benefits they may receive from others’ actions and the intentions that underlie these actions. This creates a possible tension between evaluating what an agent did versus why they did so. In adults, similar tensions have been proposed in the theoretical evolutionary model the homo moralis, which demonstrates that adults reason with a balance between self-interest and a preference for moral behavior [54-55]. Exploring whether similar reasoning exists in early childhood may provide insight into how young children consider and navigate both actions and intentions in their social decision-making.”

4. The rationale for including "willingness to play with E2" and "social bonding" as dependent variables is unclear. I understood that prize allocation to E2 measures reciprocity and liking scores assess evaluation of E2. The purpose of the remaining two measures needs explanation in either the Introduction or Methods. Clarifying this would enable deeper discussion of why reciprocity showed condition differences while the other three measures did not.

Response: We agree that this distinction was confusing. We included social bonding as a behavioral measure of social evaluation and thus have clarified this and collapsed social bonding into an element of social evaluation for conceptual purposes.

5. The Methods section remains unclear in places. For example, were questions about reciprocity, social evaluation, and social bonding counterbalanced? How was the new toy (colorful bouncy ball) introduced to children?

Response: We have now added additional information about the order of the measures, the rationale for this order, and more information about the framing of the reciprocity measure and the conclusion of the social bonding measure.

6. P-value reporting is inconsistent (e.g., p = 0.009, p < 0.01, p = .812). Please standardize the format for accurate reporting.

Response: Thank you for pointing this out. We inaccurately reported a p-value of p <0.01. It has now been corrected to be p <.001. We have corrected all other statistics to follow APA style, reporting the exact p-value unless p < .001 and using leading zeros only when the statistic is less than 0 but can exceed 1.

7. Lines 279-281 (p. 14) state: "Taken together, our results provide evidence that children considered the prosocial action, the revealed intention, and their own personal benefits." However, the experiment does not demonstrate consideration of personal benefits. Since the ball given to E2 belonged to E1, not the participant, it had no impact on the participant's personal benefits.

Response: We did not intend for this sentence to be interpreted in this way, but we now understand that it was confusing. We have reworded this to reflect that the benefit arose from the initial prosocial action. This sentence now reads as follows:

“Taken together, our results provide evidence that children considered both the benefit of the prosocial action and the later revealed intention behind the prosocial action.”

8. Lines 300-307 (p. 15) propose belief revision to explain why only reciprocity was affected by intention. However, this explanation is insufficient for why reciprocity alone was influenced. Moreover, given the partial effects observed, it is premature to conclude that children revise moral beliefs similarly to physical and social beliefs. If discussing belief revision, you must explain why intention information revised beliefs about the prosocial act but left beliefs about the agent unaffected.

Response: We agree. We have added justification for why we believe revision about the action but not the agent may occur and we have posited this more theoretically, as an area for future exploration.

9. In summary, while the authors' substantial revisions have improved the manuscript from its initial version, it remains unclear what contribution this research makes to the extensive existing literature on this topic.

Response: Thank you for your comments. We have addressed everything mentioned and believe the manuscript is tighter and stronger now. We thank you for your careful consideration.

Thank you all again for the thoughtful attention the editor and reviewers gave our manuscript over these rounds of revisions. We believe that our manuscript is theoretically and empirically stronger and sounder now thanks to these revisions.

Sincerely,

The Authors

---

## [Decision Letter · Decision Letter 2]

24 Feb 2026

When revealed after the fact, selfish intentions undermine prosocial actions in 5-year-olds

PONE-D-25-28793R2

Dear Dr. Dotson,

We’re pleased to inform you that your manuscript has been judged scientifically suitable for publication and will be formally accepted for publication once it meets all outstanding technical requirements.

Kind regards,

Asami Shinohara

Academic Editor

PLOS One

Additional Editor Comments (optional):

Reviewers' comments:

Reviewer's Responses to Questions

**Comments to the Author**

Reviewer #2: All comments have been addressed

2. Is the manuscript technically sound, and do the data support the conclusions?

Reviewer #2: (No Response)

3. Has the statistical analysis been performed appropriately and rigorously?

Reviewer #2: (No Response)

4. Have the authors made all data underlying the findings in their manuscript fully available?

Reviewer #2: (No Response)

5. Is the manuscript presented in an intelligible fashion and written in standard English?

Reviewer #2: (No Response)

Reviewer #2: (No Response)

**Do you want your identity to be public for this peer review?** For information about this choice, including consent withdrawal, please see our Privacy Policy

Reviewer #2: No

---

## [Editor Report · Acceptance letter]

PONE-D-25-28793R2

PLOS One

Dear Dr. Dotson,

I'm pleased to inform you that your manuscript has been deemed suitable for publication in PLOS One. Congratulations! Your manuscript is now being handed over to our production team.

Kind regards,

on behalf of

Dr. Asami Shinohara

Academic Editor

PLOS One